# Life in Phases: Intra- and Inter- Molecular Phase Transitions in Protein Solutions

**DOI:** 10.3390/biom9120842

**Published:** 2019-12-08

**Authors:** Vladimir N. Uversky, Alexei V. Finkelstein

**Affiliations:** 1Department of Molecular Medicine, Morsani College of Medicine, University of South Florida, Tampa, FL 33612, USA; 2Laboratory of New Methods in Biology, Institute for Biological Instrumentation, Russian Academy of Sciences, Federal Research Center “Pushchino Scientific Center for Biological Research of the Russian Academy of Sciences”, 142290 Pushchino, Moscow, Russia; 3Institute of Protein Research, Russian Academy of Sciences, 142290 Pushchino, Moscow, Russia; 4Biology Department, Lomonosov Moscow State University, 119192 Moscow, Russia; 5Bioltechnogy Department, Lomonosov Moscow State University, 142290 Pushchino, Moscow, Russia

**Keywords:** protein folding, protein structure, intrinsically-disordered protein, molten globule, secondary structure, coil, phase transition, liquid–liquid phase separation, membraneless organelle, amyloid fibril, crystal

## Abstract

Proteins, these evolutionarily-edited biological polymers, are able to undergo intramolecular and intermolecular phase transitions. Spontaneous intramolecular phase transitions define the folding of globular proteins, whereas binding-induced, intra- and inter- molecular phase transitions play a crucial role in the functionality of many intrinsically-disordered proteins. On the other hand, intermolecular phase transitions are the behind-the-scenes players in a diverse set of macrosystemic phenomena taking place in protein solutions, such as new phase nucleation in bulk, on the interface, and on the impurities, protein crystallization, protein aggregation, the formation of amyloid fibrils, and intermolecular liquid–liquid or liquid–gel phase transitions associated with the biogenesis of membraneless organelles in the cells. This review is dedicated to the systematic analysis of the phase behavior of protein molecules and their ensembles, and provides a description of the major physical principles governing intramolecular and intermolecular phase transitions in protein solutions.

## 1. Introduction

Protein structures depend on the interplay of chain conformational entropy and the sum of multiple weak interactions of different physico-chemical natures, which can be considered as “conformational forces” defining the free energy change between the folded and unfolded states that are related to protein stability. In this review, we will focus on proteins that exist mainly in the aqueous environment.

Among the weak, noncovalent interactions stabilizing protein structures are hydrogen bonds (having up to 25–40 kJ/mol in vacuum or nonpolar medium, but only 8–10 kJ/mol in aqueous environment), salt bridges (having up to 100 kJ/mol in the absence of water, but only about 5–10 kJ/mol in aqueous environment), long-range electrostatic interactions (which are weaker but more numerous than salt-bridges, and whose free energy depends on the distance between the charges and on their environment), van der Waals interactions (of about 3 kJ/mol for interaction of two methyl groups), and hydrophobic interactions (free energy of which scales with the size of the solute surface as ≈10 kJ/mol/nm^2^, which, for a methyl group with a surface area of about 1 nm^2^, would amount to ≈10 kJ/mol) [1]. Since these interactions are extremely condition dependent, the presence (or absence) of a structure in a query protein is condition-dependent too. Furthermore, due to their different physico-chemical natures, various conformational forces can differently react to changes in environmental conditions. In fact, although high concentrations of strong denaturants, such as guanidinium chloride (GdmCl), guanidinium thiocyanate (GTC), or urea can efficiently suppress all (or almost all) intramolecular conformational interactions leading to an almost complete unfolding of a globular protein into a highly-disordered, random, coil-like conformation [2,3,4,5], often, environmental alterations can decrease (or even completely eliminate) part of the conformational interactions, whereas the remaining interactions remain unchanged or even strengthen.

Therefore, although it is commonly believed that all the necessary information for a given protein to correctly fold to the specific, unique, and biologically-active conformation is included in its amino acid sequence [1,6,7,8], this, actually, only concerns distinct proteins in their physiological environment, while, in general, the crucial effect of environment should not be excluded. In fact, changes in the environment of a globular protein can cause a wide spectrum of structural changes, ranging from an almost complete unfolding in the concentrated solutions of a strong denaturant to a more subtle denaturation (which is typically is associated with the loss of both the unique 3D structure and the unique biological activity) under some “mild denaturing conditions”. In other words, the complete unfolding of a protein does not necessarily represent the only consequence of denaturation. Instead, some partially-folded conformations can possess properties that are in-between the properties of the folded and the completely unfolded states. As a result, depending on the peculiarities of their environments, the chains of globular proteins may exist in at least four different states in aqueous media, i.e., their own native (ordered) conformation, molten globule, premolten globule, and unfolded [1,9,10,11,12,13,14,15,16,17,18,19,20,21] (Figure 1), not to mention other forms that can be induced by nonaqueous environments, such as alcohols, membranes, or other proteins, as well as by post-translational modifications of their chains.

One should keep in mind that although many globular proteins possess clearly defined and unique 3D structures, these structures are rather heterogeneous, with the ordering degrees being greatly diversified in the different parts of a given protein. Such structural heterogeneity is seen in X-ray data as the variability of the values of the *B*-factor characterizing the mobility of separate atoms in a protein [23,24], with the atoms of the active center of an enzyme being typically characterized by the lowest *B*-factor. Additionally, some globular proteins have highly dynamic or even completely unstructured regions (e.g., loops and terminal fragments) that correspond to the regions of missing electron density, being therefore undetectable by X-ray analysis [25,26,27,28,29].

In addition to the “traditional” ordered proteins that “obey” classical function-structure paradigm, where a specific function of a protein is determined by its unique and rigid 3D structure encoded in a unique amino acid sequence encrypted in a corresponding gene, recently, we have witnessed an increased appreciation of intrinsically-disordered proteins, i.e., biologically-active proteins having no unique structures, at least before interactions with other molecules. In fact, it is recognized now that all organisms from all kingdoms of life (bacteria, archaea, and eukaryotes) and all viruses contain discernible levels of intrinsically-disordered proteins (IDPs) or hybrid proteins containing ordered domains and intrinsically-disordered protein regions (IDPRs) [30,31,32,33,34,35,36]. Furthermore, based on computational analyses, it has been concluded that IDPs/IDPRs universally exist in all living organisms [30,31,32,35,37,38,39,40,41,42,43,44,45,46,47,48,49,50,51], with the penetrance of disorder in different species increasing with the rise of the organism complexity [30,31,32,35,52]. Here, the expected fractions of sequences predicted to have long IDPRs (30 residues or longer) are arranged in the following order: Bacteria ~ Archaea << Eukaryotes [30,32,36,38,53]. This increase in the amount of disorder in eukaryotes was attributed to the expansion of the significance of cellular signaling that often relies on IDPs/IDPRs [33,54,55,56,57,58,59,60]. Furthermore, only a very small fraction of proteins with known crystal structures in the Protein Data Bank (PDB) is completely devoid of disorder [25,61].

The biological functions of many IDPs/IDPRs are strongly disorder-dependent, and can be described in terms of the entropic chain activities, where an extended random-coil conformation maintaining flexibility while carrying out its function is needed [34,62]. Stochastic machines can serve as illustrative examples of such dynamic signaling complexes with “entropic” chain activities [63]. Structured domains in these stochastic machines are connected by long flexible linkers [63,64,65,66]. As a result, the action of such machines does not depend on coordinated conformational changes. Instead, they operate via uncoordinated and stochastic movements of their flexible arms (long, disordered linkers), which, despite being engaged in constant and chaotic movements, can eventually enable productive functionality [63]. However, there is a large set of IDPs/IDPRs that can fold at binding to their partners [55,56,67]. Since various systems may have different degrees of such binding-induced folding, the resulting complexes are characterized by wide structural and functional heterogeneities [68,69]. The existence of such foldable IDPs was used to argue that “a partly disordered polypeptide may be capable of specific recognition through a conformer selection mechanism, but then it is the ordered population that reacts, and the disorder is neither intrinsic nor functional”, and that, therefore, such structureless proteins should be considered as “proteins waiting for a partner” (PWPs), which serve as parts of multi-component complexes that do not fold correctly in the absence of other components [70]. Although based on these arguments it was concluded that molecular disorder is not compatible with protein function [70], and although protein functionality seems to commonly originate from disorder-to-order transitions, the existence of the opposite scenario was emphasized, where the local or even global functional unfolding of ordered proteins represents an important prerequisite for their functionality [71]. This regulated unfolding [72], or dormant or conditional disorder, shows induced [73] and transient character [74]. It can be awoken by a broad range of environmental factors, such as the release of autoinhibition, light exposure, changes in redox potential, mechanical force, various posttranslational modifications, or changes in pH or temperature, or by specific interactions with ligands, membranes, nucleic acids, or other proteins [71]. It was pointed out that the existence of such dormant disorder phenomenon signifies the global importance of intrinsic disorder for protein functionality.

Since flexible protein regions are known to serve as the primary targets for the proteolytic attacks of various proteases [75,76,77,78,79,80], the fact that many IDPs and IDPRs are not cleaved by cellular proteases represents a very interesting conundrum pertaining to the peculiarities of the cellular life of intrinsically-disordered proteins and hybrid proteins with ordered domains and disordered regions. In fact, it used to be commonly believed that only folding can protect polypeptide chains against proteolysis. In line with this conjecture, it is known that IDPs are exceptionally sensitive to proteolysis in vitro [81,82,83], with this high proteolytic sensitivity being considered one of the characteristic features of IDPs/IDPRs [33,34,55,56,84,85,86]. Furthermore, for signaling IDPs, it was postulated that fast cellular degradation represents one of the mechanisms of their functional regulation [34,56,57,62,87,88,89]. However, not all IDPs/IDPRs are degraded rapidly inside the cells, and analyses of the available data on intracellular protein half-lives suggest that the presence of intrinsic disorder in a protein does not imply a significantly shorter half-life of its carrier [90,91]. One of the potential explanations for this phenomenon is based on the hypothesis that in the cellular milieu, many IDPs/IDPRs exist largely in a bound state, thereby possessing context-dependent resistance to proteolysis [91]. Several different mechanisms were proposed for such binding-induced protection against cellular degradation by default. In fact, some IDPs/IDPRs can always be bound to their specific biological partners. In such cases, binding-induced structural stabilization of IDPs might represent a “by-product” of their functional interactions. For other IDPs/IDPRs, such cellular degradation by default is prevented by interactions with specific binding partners (e.g., proteasome gate keepers and nanny proteins) that interact with IDPs solely or principally to prevent their degradation [91]. Still other IDPs/IDPRs are protected from degradation via interactions with “decoy” DNA binding sites or via intramolecular interactions that minimize the number of relatively unconstrained long IDPRs capable of accessing 20S core proteasome [91]. Importantly, despite the obvious hypothesis that chaperones may offer direct protection from cellular degradation by default by preferential binding IDPs/IDPRs in the cell, comprehensive bioinformatics analysis of chaperone-binding and nonchaperone-binding proteins in three taxonomic groups revealed that there is, in fact, a negative correlation between the intrinsic disorder propensity in proteins and their tendency to be binding partners of chaperones [81]. In other words, the preferential cellular partners of chaperones are ordered proteins, which requires more assistance for folding and protection from misfolding and aggregation than IDPs/IDPRs [81]. Among the other molecular mechanisms used to explain the protection of IDPs/IDPRs from cellular proteases are specific amino acid compositions of disordered proteins and regions and tight regulation of intracellular proteases [81]. Furthermore, the stability of proteins can be regulated by various post-translational modifications (PTMs), and it was shown that IDPs/IDPRs can serve as substrates of twice as many kinases as ordered proteins [92,93]. Since in addition to the conformational stability of proteins, PTMs can affect their activity, folding, interactions, turnover, and localization, and since IDPs/IDPRs are often serve as primary targets for the modifying enzymes [33,56,60,88,94,95,96,97,98,99], it is likely that PTMs have other roles in protecting IDPs against cellular degradation by default, e.g., by ensuring preferential localization of IDPs and hybrid proteins to the cellular compartments that do not contain proteases. An illustrative example of such compartmentalization is given by some proteinaceous membraneless organelles (PMLOs), where IDPs/IDPRs can be protected from the proteolytic degradation by the preferential protease exclusion from these organelles [100]. In conclusion, although several potential explanations for the remarkable IDP/IDPR protection against cellular degradation by default were given, it seems that there is no universal mechanism for such protection, indicating that protein degradation is not determined by a single characteristic, such as intrinsic disorder propensity, representing a complex, multifactorial process with prominent protein-to-protein variations.

An important feature of a polypeptide chain is its ability to undergo intramolecular and intermolecular phase transitions. The discovery that denaturation of the globular, native, rigid, one-domain protein structure can occur as an “all-or-none” transition [101,102], i.e., a transition without accumulation of visible intermediates, is one of the cornerstones of protein physics. In this article, we consider several aspects of such phase transitions, paying special attention to the spontaneous intramolecular phase transitions in the solutions of ordered proteins undergoing either equilibrium “native state—molten globule—unfolded chain” conformational transitions or kinetic “native state ↔ unfolded chain” structural transitions, induced intramolecular phase transitions in the solutions of IDPs promoted by their interactions with binding partners, transitions associated with the nucleation in bulk, on the interface, and on the impurities, as well as macrosystemic phenomena of protein aggregation, the formation of amyloid fibrils, and protein crystallization, and intermolecular liquid–liquid or liquid–gel phase transitions that are associated with the biogenesis of membraneless organelles in the cells.

## 2. Intramolecular Phase Transitions of Ordered Proteins

### 2.1. Brief Description of Major Partially-Folded States of Globular Proteins

#### 2.1.1. Molten Globule

The structural properties of a protein molecule in the molten globule are well known, and have been outlined in a number of reviews (e.g., [12,13,79,103,104,105,106,107,108,109,110,111,112,113,114,115,116,117]). Although the molten globular protein completely lacks a rigid, cooperatively-melting, tertiary structure, or has only a trace of such a structure, i.e., is denatured, it preserves high levels of a native-like, secondary structure [12,13,79,103,104,105,106,107,108,109,110,111,112,113,114,115,116,117,118]. Molten globules are compact (in comparison with the native state, their hydrodynamic radii are increased by less than 15%, which translates into a ~50% increase in volume) [12,13,18,21,79,103,104,105,106,107,108,109,110,111,112,113,114,115,116,117,119] and, as evidenced by small-angle X-ray scattering, have a globular structure, which is usually similar to that of native globular proteins [120,121,122,123,124]. The analysis of the molten globules of several proteins by solution NMR spectroscopy revealed that a protein molecule in this intermediate state does not only have a native-like secondary structure, but that it also shows a native-like folding pattern [114,125,126,127,128,129,130,131,132,133,134,135]. This is not surprising, because what is good for the molten globule is probably good for a well-folded protein structure as well. However, there are interesting exceptions; the best known one concerns the molten globule of β-lactoglobulin [136]. Although the native protein is a β-structural molecule, its folding intermediate contains nonnative α-helices, according to the CD spectra.

Despite their secondary-structure-enriched, relatively compact, and globular conformations, molten globules are characterized by the considerable increase in the accessibility of a protein molecule to proteases [75,76,77,78,79,80]. Finally, one of the most characteristic features of the molten globule is its high affinity to the hydrophobic fluorescence probes (such as 8-anilinonaphthalene-1-sulfonate, ANS or 1,1′-bis(4-anilino-5-naphthalenesulfonic acid), bis-ANS) [137,138,139,140].

#### 2.1.2. Premolten Globule

Similar to molten globules, the premolten, globular, partially-folded intermediate represents a denatured conformation with no rigid tertiary structure. Premolten globules are markedly less compact than the molten globular or native states of a protein with a given molecular mass, although these intermediates are still noticeably more compact than random coils. In fact, compared to the native state, the hydrodynamic volumes of the molten globule, premolten globule, and unfolded states are increased 1.5-, ~3-, and ~12-fold, respectively. However, there is no globular structure in a premolten globular protein [15,124], suggesting that this conformation is likely a partially-ordered form of a “squeezed” coil. In line with this hypothesis, premolten globular protein is characterized by the preservation of considerable levels of secondary structure. However, this residual ordering (protein molecule in the premolten globule state usually has ~50% or even less of native secondary structure) is much less pronounced than that of the molten globule, which typically shows a native-like secondary structure. Finally, at least part of the solvent-accessible hydrophobic clusters is already formed in the premolten globule, as evidenced by the ability of this intermediate to interact with the hydrophobic fluorescent probe ANS [11,13,14,15,17,116].

Finally, it has been shown that the premolten globule (with a relatively large secondary structure content), as well as the unfolded state (with a low content of residual secondary structure), is separated from the molten globule by a sharp transition, which, in some proteins, represents an “all-or-none” transition, i.e., an intramolecular analog of the first order phase transition [11,13,14,15,17,116,141]. This means that in these cases, the molten and premolten globules represent different thermodynamic (phase) states of a polypeptide chain [10,141]. It seems that the aforementioned all-or-none transition is due to the formation of a secondary structure within a swollen premolten globule [142], especially since it is known that a β-sheet formation is of an “all-or-none” kind [143,144,145]. No sharp transition from the premolten state to the random coil has been reported as of yet.

### 2.2. Thermodynamics of the Protein Denaturation. “Wet” and “Dry” Molten Globules

Before the molten globule state was discovered, protein denaturation was typically thought of as the complete decay of the unique protein structure, i.e., a transition to the coil. After the theoretical prediction [22], and then the experimental discovery of the molten globule [146,147], it became clear that the denatured protein can be rather dense, nearly as much as that of the native protein, as well as loose, like the coil, depending on the solvent’s strength and the hydrophobicity of the protein chain.

To understand the molecular basis of protein denaturation, one has to explain why two two equally stable phase states of the protein chain can exist, and why they are separated by a free energy barrier (which is required for an “all-or-none” transition). In other words, one has to explain why the protein globule cannot decay by gradual (barrierless [148] or overcoming a very low free-energy barrier [149]) swelling, as typical polymers do due to the persistent connectivity of their chains [150].

In so doing [151,152,153], one has to consider the major characteristics of proteins defining their difference from “normal” polymers: (i) each globular protein possesses the only chain fold with a peculiar stability; (ii) flexible side groups are linked to a much more rigid protein-chain backbone; and (iii) the packing of a native globular protein is as tight as the packing of a molecular crystal (although with no crystal lattice), where the van der Waals volumes of atoms occupy 70–80% of the volume, whereas only 60–65% of the volume in liquids (melts) is occupied by the van der Waals volumes of atoms [154].

The side chains of the protein can undergo a rotational isomerization. This is done by jumps between the allowed conformations of the side chains. Each jump necessitates some vacant volume near the side chain that jumps. However, since the native protein structure has a tight packing of the chain (which contributes to the enhanced stability of this fold), each jump needs some extra free volume for landing (see inset in Figure 2).

Note that the flexible side groups sit at the rigid backbone. The backbone is especially rigid inside the globule, where the α- and β-structures hide H-bonds of their polar peptide groups from the dense hydrophobic environment, and these α- and β-structures are stable, at least until water molecules penetrate into the globule (which requires about the same free volume as the side chain jumps). Therefore, the free volume can be hardly made for a separate jumping side chain, and each of the rigid secondary structure elements, with the entire forest of flexible side chains attached, moves as a whole (at least at the very beginning of the globule’s expansion). Therefore, the expansion of the closely-packed globule, carried out by the moving apart of the rigid α- and β-structures, creates about the same amount of free space near each side group; these spaces are either insufficient for the isomerization of each of the side groups (when the globule expansion is still too small), or are already sufficient for the isomerization of many of them. This means that liberation of the side groups (as well as water penetration) can occur only when the globule expansion crosses a particular threshold, i.e., the “barrier”.

Analysis of the properties of a protein globule at different levels of its uniform expansion [151,152,153] shows that an expanded state of the protein globule can be as stable as its native (solid) state, but only after the density barrier has been passed. (It should be noted here that this analysis of a uniform globule’s expansion, illustrated by Figure 2, does not aim to model the protein unfolding kinetics, which occurs via intramolecular separation of the native and denatured phases, as shown in Figure 3a below).

Thus, a small expansion of the compact native protein globule is *always* unfavorable [151,152,153], because it *already* increases the globule’s energy (whose parts already lose their close packing), but does not yet increase the globule’s entropy (since it does not yet liberate the rotational isomerization of the side groups) or allow entry of water into the protein core. That is, the globule’s free energy always increases with a small expansion. In contrast, a large globule’s expansion liberates the rotational isomerization of the side groups and leads (at high enough temperature) to a decrease of the free energy. As a result, protein denaturation occurs not gradually, but as a jump over the free energy barrier, leading to the “all-or-none” kind of transition (Figure 2).

The aforementioned mechanism is related to the transition of a native globular state to any denatured form: molten globule, premolten globule, or coil [141,152]. Therefore, the protein structure tolerates, without significant change, a change of ambient conditions up to a certain limit, and then melts as a whole, like a macroscopic crystal. This provides the reliability of its biological functioning. Put differently, a sudden jump in entropy (mainly entropy of the side chains), which may happen only after the expansion of the globule crosses a particular threshold, explains the origin of the “all-or-none” transition separating the native and denatured state. Such a global entropy jump happens because of the fact that the side chains cannot be liberated one-by-one, since they are held by rigid backbone that coordinates their positions.

The pores in the molten globule are usually “wet,” that is, they are occupied by the solvent, because a water molecule inside the protein is still better than a vacuum [151,152,153]. Experimentally, the “wetness” of the molten globule is proven by the absence of a visible decrease in the protein floating density [155] after denaturation of any kind.

When the solvent sticks to the protein core (consisting mainly of hydrophobic groups) not too strongly, it only occupies the pores that have been already formed in the molten globule core to ensure side-chain movements, but it does not expand the globule (just as water does not expand a sponge, although it occupies its pores), and does not make new pores. Then, the denatured protein remains the wet molten globule [152,155].

Like the “wet” molten globule, a “dry” molten globule (having no water in its pores) was predicted theoretically [152]; an analysis showed that the dry molten globule should be less stable than the wet one, and therefore, that it is hardly suited to playing the role of a stable, accumulating intermediate in protein melting. However, it has been found [156] that the dry molten globule emerges during fluctuations preceding protein melting.

The molten globule compactness is maintained by the residual hydrophobic interactions of its side groups. They were found to be not very strong. Even in the apomyoglobin, the molten globule (which has a well-developed secondary structure and almost native chain topology of packing of the most of its chain [157]), the residual interactions between the hydrophobic residues appear to be three or four times weaker than those in the native protein [158]; these residual interactions are entirely missing for some hydrophobic residues (which accentuates the heterogeneity of the molten globule; see [13]).

If the residual hydrophobic interactions are weak, i.e., if either the hydrophobicity of the chain is low or the protein chain strongly attract the solvent, the solvent starts to expand the pores, and the globule swells. The greater the attraction between the solvent and the protein chain, and the smaller the attraction within the protein chain, the greater the chain swelling, leading to the transition to the premolten globule and then to the random coil.

### 2.3. Kinetics of the “Unfolded Chain ↔ Native State” Transitions

The ability of polypeptide chains of globular proteins to spontaneously form their spatial structures is a long-standing puzzle in molecular biology. Numerous pieces of evidence (such as the independence of the protein structures folded both in vivo and in vitro on the initial states and configurations of the chains) show that the native protein structure is the most stable of all structures of the chain under physiological conditions [6,7,8]. Here, it is worth noting that experiments have shown that there is no fundamental difference between the in vivo (cotranslational) folding [159,160,161] and in vitro folding of truncated and complete chains [162], at least for small proteins; in both cases, native-like structures emerge only after the entire sequence is available.

The experimentally-measured folding times range from microseconds for small to hours for large single-domain globular proteins; the difference (about 10 orders of magnitude) is the same as that between the life span of a mosquito and the age of the Universe. But these microseconds or even hours are negligible compared to the time necessary to iterate over all possible structures of the protein chain and to find the most stable of them; this requires something like 3*^L^* or even 10*^L^* picoseconds (where *L*, usually ~100, is the number of amino acid residues in the protein chain), i.e., billions of years [163,164]. Consideration of this “Levinthal’s paradox” led to the idea that the energy landscapes of protein chains must be somehow inclined, like funnels, towards the native protein structures; this would facilitate a sequential protein folding [165,166,167]. Landscapes of this kind can drastically decrease the time required for the protein chain folding by reducing, for these chains, the free-energy barrier, which, ensuring an “all-or-none” transition, separates their unfolded (U) and natively-folded (N) states. It is noteworthy that the energy landscape is, on average, automatically inclined towards the most stable protein structure, because the interactions present in this structure are, on average, stronger than the other ones. As a result, native-like folding intermediates (possessing a part of these native interactions) are, on average, more stable than the “nonnative-like” folds that do not possess them. In line with earlier analytical estimates [168], computer experiments have shown that a model polymer whose random sequence was slightly “edited” to make the free energy of its most stable fold lower than that of any other fold by at least by a few kcal/mol [169,170] finds this most stable fold in a time, which is many orders of magnitude smaller than the time necessary to iterate over all the possible chain structures.

A physical theory that not only solved the Levinthal’s paradox, but also estimated the dependence of the protein folding time on protein size and shape, was first presented in the second half of 1990s [171,172,173]. This theory considers overcoming the free-energy barrier separating the natively-folded (N) and unfolded (U) states of protein chains. This barrier occurs in both the uniform (Figure 2) and nonuniform (Figure 3a) expansions of the globule, but the height of the former barrier is proportional to the protein size, while the height of the latter (occurring via intramolecular separation of the native and denatured phases) is much lower, being proportional to the size of the globule cross-section, i.e., the protein chain length to the power of 2/3. Therefore, the main pathway of the N ↔ U transition goes via just this lower barrier, the essence of which is an intramolecular phase separation.

The developed theory is applicable to protein and “protein-like” sequences, i.e., those having a distinguished chain fold, the free energy of which is lower than that of any other fold by at least a few *k_B_T_melt_* [169,170,174,175] (where *T_melt_* is its melting temperature). In this theory, a special role is played by the point of thermodynamic (and thus kinetic) equilibrium between the N and U states. 

Here, the theory obtains the simplest form, because both halves (the native-like and unfolded) of a semi-folded protein have equal free energies, so that the free energy of semi-folded protein is only determined by the interface between these two halves (that is, mainly by the surface free energy of the “native phase”). The maximal unavoidable interface between the N and U states occurring in the course on U ↔ N transition includes ≈*L*^2/3^ amino acid residues (*L* being the number of residues in a protein chain). Therefore, the barrier heights also scale with the protein size as ≈*L*^2/3^, and, therefore, the corresponding protein folding time scales as ~exp (≈*L*^2/3^), rather than as ~exp (≈*L*), appearing in the Levinthal’s paradox. This scaling, ~exp (≈*L*^2/3^), means that the protein folding time is many orders of magnitude less than the time ~exp (≈*L*), which is necessary to iterate over all possible chain structures.

A theoretical estimate of the folding time is based on the conventional transition state theory [177,178,179]. For the N↔U equilibrium, an accurate estimate of the folding (and unfolding) time for a protein chain of *L* amino acid residues gives
TIME ∼ τ × exp[(0.5 ÷ 1.5)*L*^2/3^],(1)where τ ≈ 10 ns is the time of the conformational rearrangement of one residue (measured for the helix ↔ coli transition) [180].

The lower estimate (TIME ∼ τ × exp[0.5*L*^2/3^]) corresponds to the proteins with “simple” chain folds, which have a transition state (“folding nucleus”) structure where the N–U interface is not covered by the closed unfolded loops; the energy loss for one residue of the phase surface, ≈0.5 *k_B_T_melt_*, is taken as ε/4, where ε ≈ 1.3 kcal/mol ≈ 2*k_B_T_melt_* is the average heat of native protein melting per residue [102] (this is the first empirical parameter used by the theory, while τ ≈ 10 ns is the second and the last used empirical parameter).

The upper estimate (TIME ∼ τ × exp[1.5*L*^2/3^]) corresponds to the proteins with “complicated” chain folds, which have a transition state (“folding nucleus”) structure where the N–U interface is maximally covered by the closed unfolded loops. Strictly speaking, this upper estimate is TIME ∼ τ × exp[(0.5 + ^5^/_12_ln(3*L*^1/3^))*L*^2/3^], where the logarithmic term follows [171,181] from averaging the Flory’s estimate for the entropy of closed loops, but for protein chains of a normal size, ~50 ÷ 200 residues, this ^5^/_12_ln(3*L*^1/3^) is so close to 1 that there is no need to overcomplicate the simple result given by Equation (1).

Theories developed for the prediction of protein folding nuclei, experimentally studied by Alan Fersht [182] and others that one can find in [183,184,185,186,187,188,189,190,191,192,193], and a more detailed theoretical consideration of folding times for proteins of different sizes, chain folds, and stabilities, are given in [176,181,194,195,196,197]. A limited-influence chain knotting and the SS-bonds in the single-domain proteins on the folding rate were estimated in [173,187], and the influence of the native structure stability on the folding rate was estimated in [176] (see also [198]).

The aforementioned estimate (1) of protein folding rates, obtained in 1997, was confirmed by the subsequently obtained experimental data [176,194]; see Figure 3b.

However, one can see that the derived theory of protein folding rates explains Levinthal’s paradox “in *non*-Levinthal’s terms”, i.e., it deals with phase separation and free energy barriers, but gives no estimate as to the number of structures to be iterated over in a search for the most stable chain fold, and offers no explanation as to why such an iteration is feasible, at least for small globular proteins (or domains) of ~100 amino acid residues.

Our answer is that the Levinthal’s paradox assumed that the search should be done among *all* conformations of the protein chain (which is indeed impossible), while the search among low-energy folds *only* (i.e., only among compact and well-structured globules), which is done at the level of protein secondary structure assembly (Figure 4), is by many orders of magnitude less voluminous, and is therefore, feasible. A rough estimate [181,199,200] leads to the conclusion that at the level of secondary structure assemblies (or, in other words, at the level of potential molten globules), the search volume does not exceed
~*L^N^* ~ exp[^1^/_4_ln(*L*) *L*^2/3^](2)for a protein chain of *L* amino acid residues and *N* secondary structure elements, which, in the main term, scales approximately as the exponent in the aforementioned upper estimate (τ × exp[(0.5 + ^5^/_12_ln(3*L*^1/3^)) *L*^2/3^] ≈ τ × exp[1.5*L*^2/3^]) of the protein folding time.

## 3. Intramolecular Phase Transitions in Disordered Proteins Induced by Interactions with Binding Partners

The interplay between the amino acid sequence of a protein and environment defines the ability of a polypeptide chain to fold, misfold, or be intrinsically disordered. Although one can induce different degrees of disorder in a molecule of a globular protein by changes in its environment (to generate molten globule, premolten globule, and coil-like states [11,12,13,14,16,17,201]), IDPs/IDPRs can be differently disordered under the same physiological conditions and exist as highly-dynamic, conformational ensembles of collapsed (native molten globules) or extended disordered species (native premolten globules and native coils) [17,33,202]. In other words, contrarily to globular proteins that have unique 3D structures under physiological conditions, IDPs/IDPRs exist under the same conditions as dynamic, conformational ensembles with quite different structures that interconvert on a number of timescales. Accumulated data on the structural heterogeneity of IDPs suggest that the representation of these proteins as members of three well-defined structural classes (native molten globules, native premolten globules, and native coils) is an oversimplification. In fact, IDPs/IDPRs might contain foldons (i.e., independently foldable protein units, which should not be mixed with domains, since single-domain proteins might have several foldons [203,204,205,206,207], as was shown for cytochrome *c* [208], apo-cytochrome *b*_562_, ribonuclease H, dimeric triosephophate isomerase, the OspA protein of Borrelia [203], and staphylococcal nuclease [209]), inducible foldons (which are IDPRs capable of at least partial folding, promoted by their interactions with binding partners), morphing inducible foldons (IDPRs with the potential to fold differently due to binding to different partners), semi-foldons (regions that are always in a semi-folded form), and nonfoldons (IDPRs that never fold). On the other hand, the functionality of many ordered proteins depends on the presence of ‘unfoldons’, i.e., regions of ordered proteins that undergo order-to-disorder transitions to make proteins active [210]. Therefore, based on currently available data, one can conclude that intrinsic disorder can have multiple faces, and can affect different levels of protein structural organization, where either whole protein or various regions are disordered to a different degree. Based on these considerations, it has been proposed that functional proteins represent a continuous spectrum of differently-structured/disordered conformations that ranges from fully ordered to completely structureless species and everything in between [210]. It was also pointed out that no boundary is present between the ordered proteins and IDPs. Instead, the structure–disorder space of a protein represents a continuum [210] that defines the protein structure-function continuum [211,212,213,214], where instead of the classical “one gene—one protein—one structure—one function” model, any protein represents a dynamic conformational ensemble containing multiple conformational/basic, inducible/modified, and functioning proteoforms. Proteoforms represent a set of distinct protein molecules encoded by a single gene. They originate from allelic variations and various pretranslational mechanisms affecting genes, such as the production of multiple mRNA variants by alternative splicing and mRNA editing. They can also be generated by numerous changes induced in the chemical structures of proteins by various post-translational modifications (PTMs) [215]. Also, some of them can be linked to the presence of IDPRs, or can originate from the functionality [211]. Therefore, the protein structure–function continuum suggests that any protein can be characterized by a broad spectrum of structural features, and can possess various functional potentials [211,212,216]. As a result, IDPs/IDPRs are not homogeneous, but represent a very complex mixture of potentially foldable, partially foldable, differently foldable, or not foldable segments [210,217]. In other words, IDPs/IDPRs behave as highly frustrated systems with no single folded state. This is reflected in their free energy landscapes, which are relatively flat and simple, and do not have a deep energy minimum seen in the free energy landscape of the ordered globular protein, representing instead a kind of ‘hilly plateau’, where hills correspond to forbidden conformations [16,218,219]. Such a simplified and flattened energy landscape is extremely sensitive to different environmental changes that can modify the landscape in a number of different ways, making some energy minima deeper and some energy barriers higher. This explains the conformational plasticity of IDPs/IDPRs, their extreme sensitivity to changes in the environment, and their ability to specifically interact with many partners of different natures and to fold differently as a result of these interactions [210].

The lack of rigid structures in the IDPs/IDPRs is encoded in the specific features of their amino acid sequences, such as, for proteins/regions with extended disorder, the presence of numerous uncompensated charged groups (often negative) giving rise to their high net charges at neutral pH and extreme pI values [220,221,222], and a low content of hydrophobic residues [220,221]. On a more global level, amino acid sequences of IDPs/IDPRs have several common features [223,224], such as depletion in the order-promoting residues that would normally form the hydrophobic core of a folded globular protein (e.g., bulky hydrophobic (Ile, Leu, and Val) and aromatic (Trp, Tyr, and Phe) amino acids) and Cys residues. On the other hand, IDPs/IDPRs are noticeably enriched in disorder-promoting amino acids, such as polar residues Arg, Gln, Ser, Glu, and Lys, as well as Gly, Ala, and a hydrophobic structure-breaker Pro [33,225,226,227,228]. However, one should keep in mind that although being generally depleted in hydrophobic residues, IDPs/IDPRs still contain some strategically-placed hydrophobic residues, which could be of crucial functional importance. In fact, similar to ordered proteins containing characteristic patterns of hydrophobic and hydrophilic residues which are important for protein folding and function, amino acid sequences of IDPs/IDPRs are also patterned, and these proteins are known to contain so-called molecular recognition features, i.e., regions which are disordered in the unbound form but which can at least partially fold upon interacting with specific partners [55,56,67]. Importantly, since the degree of such binding-induced folding is different for different proteins, the resulting complexes are characterized by broad structural and functional heterogeneity [68,69]. Due to their ability to fill the gaps and cracks between the structural elements of a binding partner [229], IDPs/IDPRs can act as molecular glue or mortar [56]. Since interaction with partners can initiate at least partial conjoint binding-induced folding, IDPs/IDPRs can also serve as molecular epoxy [230,231,232]. The dynamic ‘on–off’ switch-type interactions commonly found in signaling networks are dependent on intrinsic disorder, since the ability to bind partners with high specificity and low affinity represents one of the specific features of disorder-based interactions [34,233,234]. Many IDPs/IDPRs serve as morphing shape-changers that are able to differently fold as a result of binding to different partners [17,67,87,235,236,237], with the binding regions of such morphing IDPs/IDPRs being able to adopt completely different structures upon binding to the divergent partners [58,65,238,239,240].

Under physiological conditions, the capability of a globular protein to gain ordered structure is encoded in its amino acid sequence that contains, so to say, a “blueprint” of a final structure. This “blueprint” can be complete; then, the proteins are foldable, and they fold spontaneously without help from external factors [241,242,243]. The facts that IDPs/IDPRs cannot spontaneously fold into unique 3D structures and that interactions with specific partners can resolve their foldability problem indicate that some parts of their “blueprints” are missing, and that these missed parts are provided by the binding partners. Although not quite literally, this binding-induced folding of IDPs/IDPRs can be approximated by the interaction-promoted folding and assembly of a globular protein from its polypeptide fragments, as was shown for Trp repressor [244], SH2 domain [245], maltose binding protein [246], oxyanion-translocating ATPase [247], barnase [248], rhodopsin [249,250], B1 domain of streptococcal protein G [251], pig heart CoA transferase [252], *E. coli* thioredoxin [253], bacteriorhodopsin [254], G protein-coupled receptors [255], ubiquitin [256], and *E. coli* aspartate transcarbamoylase (ATCase) [257], to name a few. Curiously, this high efficiency of the functional structure restoration from the peptide fragments prompted Johnsson and Varshavsky to design a ubiquitin split protein sensor (USPS) in order to detect protein–protein interactions in vivo [258]. Here, N- and C-terminal domains of ubiquitin are fused into two proteins, the interactions of which trigger the folding of rationally designed fragments to a functional ubiquitin. This approach was further enhanced by the development of various split reporter proteins which are commonly utilized nowadays in studies of protein–protein interactions, protein localization, intracellular protein dynamics, and protein activity in living cells and animals [259]. Among the split reporter systems used in protein-fragment complementation assays are constructs based on split dihydrofolate reductase (DHFR) [260], β-galactosidase [261], green fluorescent protein (GFP) [262], firefly and renilla luciferase [263], and β-lactamase [264]. We give these examples here to illustrate the idea of the “blueprint” complementation, where a functional protein with unique structure is produced from inactive fragments as a result of conjoint folding–binding events.

## 4. Nucleation in Bulk, on the Interface and on the Impurities

In the first order phase transitions like melting or crystallization, as well as in their microscopic analogs, intramolecular “all-or-none” transitions, a key role is played by the nucleation of the new phase [265,266]. Nucleation can be 3-dimensional (that is, in bulk) or 2-dimensional (on the surface or interface).

The free energy of an emerging piece of a new phase consisting of *n* > 1 particles can be estimated as(3)$$G_{3}\left(n\right)\approx n\Delta \mu +n^{\frac{2}{3}}B_{3}$$for the 3-dimensional case, and(4)$$G_{2}\left(n\right)\approx n\Delta \mu +n^{\frac{1}{2}}B_{2}$$for the 2-dimensional case [266,267,268] (Figure 5). Here, Δ*μ* < 0 is the chemical potential decrease for the molecule of the “new” phase as compared to the “old” one; $B_{3}$ > 0 and $B_{2}$ > 0 stand for the additional free energy of a molecule at the 3- and 2-dimensional phase interfaces.

The free energy of the nucleus (the unavoidable highest-free-energy structure at the pathway of growth of a piece of the emerging new phase) is obtained from equation dG(n)/dn=0, and, in the 3-dimensional case, is(5)$$G_{3}^{\#}=\frac{4B_{3}}{27}{\left[\frac{B_{3}}{-\Delta \mu }\right]}^{2}.$$

$G_{3}^{\#}$ is achieved at(6)$$n_{3}^{\#}=\frac{8}{27}{\left[\frac{B_{3}}{-\Delta \mu }\right]}^{3},$$while the “seed”, i.e., the smallest stable piece of the emerging new phase (satisfying the equation $G_{3}\left(n\right)=0$ at *n* > 1), includes(7)$$n_{3}^{0}={\left[\frac{B_{3}}{-\Delta \mu }\right]}^{3}=\frac{27}{8}n_{3}^{\#}$$particles. Similar relationships can be obtained for the 2-dimensional case.

A few interesting consequences follow from the above (cf. Figure 5) relationships:(1)At −Δμ→0, $G_{3}^{\#}\to +\infty$ and $G_{2}^{\#}\to +\infty$, which, according to conventional transition state theory, means that the time of the first order phase transition (exponentially dependent on the $G^{\#}$ value) is infinitely high near the point of thermodynamic equilibrium of the “new” and “old” macroscopic phases. This is a kinetic origin of hysteresis, overcooled liquids, etc., and, by the way, of the enormous time required for the formation of β-sheets in long polypeptides [144,145]. It is worth mentioning that extremely slow nucleation leads to the formation of single and extremely large compact pieces of the erasing phase.(2)There is a kind of competition between in bulk and on-surface nucleation of the new phase. At −Δμ→0, $G_{3}^{\#}$ turns to infinity in proportion to ${\left[\frac{B_{3}}{-\Delta \mu }\right]}^{2}$, while $G_{2}^{\#}$ turns to infinity in proportion to only $\left[\frac{B_{2}}{-\Delta \mu }\right]$, i.e., much more slowly. This means that close to the conditions of phase equilibrium, 3-dimensional (“in bulk”) nucleation becomes kinetically impossible due to the very large ${\left[\frac{B_{3}}{-\Delta \mu }\right]}^{2}$ value, while the 2-dimensional (“on surface”) nucleation can still avoid kinetic problems, and occurs until $\left[\frac{B_{2}}{-\Delta \mu }\right]$ also becomes too large.(3)In contrast, when the phases are far from the equilibrium, that is −Δμ increases and starts to approach $B_{3}$ and $B_{2}$ (or $B_{3}$ and $B_{2}$ are small and approach −Δμ), the nucleation in bulk should become fast and overcome the on-surface nucleation, because the surface layer is several orders of magnitude smaller than the bulk. It is worth mentioning that fast nucleation leads to the formation of many pieces of the erasing phase that can glue together, forming noncompact, amorphous, or branched aggregates.(4)If an all-or-none transition occurs in a microscopic body that includes *L* particles only, the “new” phase can be stable only if the seed of the arising phase is smaller than *L*, i.e., $n_{3}^{0}={\left[\frac{B_{3}}{-\Delta \mu }\right]}^{3}\leq L$. This means that the new phase can arise only when its stability exceeds some threshold, i.e., $-\Delta \mu \geq -\Delta \mu_{L}\equiv \frac{B_{3}}{L^{1/3}}$. At the mid-transition point, where both phases have equal stability, and thus $-\Delta \mu =\frac{B_{3}}{L^{1/3}}$, the transition state free energy is $G_{3}^{\#}=\frac{4}{27}B_{3}L^{2/3}$, and it includes $n_{3}^{\#}=\frac{8}{27}L$ particles. This means that the time of transition to the new phase (which is as stable as that of old one) scales with *L* in the way given by Equation (1) for formation of the “native phase” of a protein (a microscopic body!), and that the folding nucleus of the new phase includes nearly ^1^/_3_ of the body, i.e., it is not small.(5)If the new phase is a little more stable than the old one, that is $-\Delta \mu =-\Delta \mu_{0}\left(1+\delta \right)$, where 0<δ≪1, the free energy of the completely formed new phase is $\Delta G_{3}\left(L,\delta \right)=-\delta \cdot B_{3}L^{2/3}$ < 0, and the transition state free energy $G_{3}^{\#}$ of nucleation of this stable phase by $\approx \frac{8}{27}\Delta G_{3}\left(L,\delta \right)$ is lower than the transition state free energy at the mid-transition point. Such an estimate has been used in [176] to describe the decrease in the protein folding time with the increase in protein stability.(6)If the new phase is formed around some local “impurity” and interacts with it with the free energy $G_{0}<0$, the free energy of the emerging phase obtains the form $G_{3}\left(n\right)\approx G_{0}+n\Delta \mu +n^{\frac{2}{3}}B_{3}$ (and $G_{2}\left(n\right)\approx G_{0}+n\Delta \mu +n^{\frac{1}{2}}B_{2}$), instead of that given by Equations (3) and (4). This correspondingly (by $G_{0}<0$) decreases the nucleation free energy $G_{3}^{\#}$ (as well as $G_{2}^{\#}$) of the new phase as compared to that given by Equation (5), and does not change the size $n_{3}^{\#}$ (as well as $n_{2}^{\#}$) of the critical nucleus given by Equation (6), but decreases the size $n_{3}^{0}$ (as well as $n_{2}^{0}$) of the “seed” relatively to that given by Equation (7).

## 5. Protein Crystallization, Amorphous Aggregation, and Fibrillation as Intermolecular Phase Transitions

In addition to the spontaneous and binding-induced intramolecular phase transitions described in the previous sections, proteins are able to undergo macrosystemic assembly processes of amorphous aggregation, fibrillation, gelation, crystallization, and liquid–liquid phase separation (see Figure 6). In these cases, insoluble protein ensembles with different degrees of packing orders are formed in protein solutions as a result of proteins undergoing “soluble–insoluble” changes accompanied by intermolecular phase transitions.

These processes are also different in terms of the degree of structural distortions induced by the environment in a protein molecule that triggers the corresponding transitions. They range from minimal structural changes in crystallization to moderate structural perturbations (denaturation) in amorphous aggregates, and to the large-scale conformational alterations that are typically required for fibrillation.

### 5.1. Protein Crystallization as a Peculiar Case of Phase Separation of Supersaturated Protein Solutions

An interesting peculiarity of the polypeptide chain of any well-structured globular protein is that its amino acid sequence guarantees the existence of the free energy barrier between the native and denatured (unfolded or partially folded) states [1,102]. This is of great importance for proper protein functioning, as the presence of such a barrier assures the structural identity of native proteins. The ability of native globular proteins to form crystals (known from Hoppe-Zeiler’s works of the 1860s, in which the author describes the method by which crystals of hemoglobin were obtained [270]) is one of the major pieces of evidence supporting this hypothesis. Protein crystallization is a consequence of protein association governed by the details of the protein structure and the peculiarities of its environment. It represents a phase transition leading to the separation of a solid phase (protein crystal) from a supersaturated protein solution. However, protein crystals are not “dry” (they, in fact, have high solvent contents, i.e., ranging from 27% to 65%, with an average of 43% [271,272]).

A supersaturated protein solution is a metastable system that, by subtle changes in the environment, can be triggered to also undergo liquid–liquid phase separation, gelation, crystallization, or aggregation [273,274]. An important note here is that the aforementioned phenomena of liquid–liquid phase separation, gelation, crystallization, or aggregation occur in the supersaturated solutions of normally-folded proteins with (almost) unperturbed 3D structures. However, since gelation and aggregation commonly come about during protein crystallization, the resulting gels and aggregates are typically considered as disordered phases [274].

Crystallization takes place in supersaturated protein solutions within a rather narrow set of conditions known as a crystallization slot, where the protein solution is characterized by specific molecular interactions involving both solvent and solute molecules, and where protein self-interactions are defined by a specific range of the osmotic second virial coefficient, B_22_ [275,276,277]. Liquid–liquid phase transition is believed to be driven by the short-range nature of protein interactions [274]. This process can serve as an illustrative example of a spinodal demixing, which represents a transition from one totally unstable thermodynamic phase (supersaturated protein solution) to two coexisting stable or metastable phases (in this case, a liquid relatively depleted in protein and a liquid rich in protein), both containing significant levels of solvent. Furthermore, because of the presence of the high protein concentration phase (when it is metastable in relation to crystallization), crystallization occurs much more rapidly in the concentrated phase-separated state than in the initial solution [273].

It has been pointed out that the aggregation and crystallization in the supersaturated solutions of globular protein can be associated with the universal behavior of the concentration fluctuations taking place in the vicinity of the “spinodal line” in the (*T*, *c*) plane (where *T* is the temperature of the system and *c* is the protein concentration) [278,279]. This spinodal line represents a boundary showing the limits of the thermodynamic stability of a homogeneous fluid, or, in other words, the line separating a region of instability of the solution against spontaneous, nonnucleated demixing [280], where (unlike nucleation) the interface free energy plays no role. In the proximity of the spinodal line, the critical divergence of amplitudes and the lifetimes of the spontaneous fluctuations of solute concentrations within the stability region occur, and can be viewed as transient demixing [280]. These divergences follow the universal scaling law describing the temperature dependence of parameter *ε*, that “measures the normalized distance of the representative point of the solution from the spinodal line in a form *ε* = (*T* − *T_s_*)/*T_s_*, where *T* is the actual temperature of the system and *T_s_* is the spinodal temperature” [278,279,280]. It has also been pointed out that light-scattering experiments in slow temperature scans can provide important information on the existence of anomalous fluctuations, temperature range, where their divergences follow the universal law, and spinodal temperature values [281]. Importantly, since the parameter *ε* reflects the global effects of all system parameters (such as additives, buffer, concentrations, pH, salts, temperature, etc.) on the solution stability, very different combinations of system parameters can give the same *ε* value.

A careful analysis of the crystal nucleation of several proteins under a variety of conditions reveals that for all the systems studied, dependencies of the nucleation rates on the *ε* parameter follow the same universal curve (or “master curve”) that covers a span of several orders of magnitude of nucleation rates [278,279]. The existence of such a master curve represents a direct consequence of the existence of the related critical concentration fluctuations and the universal divergence properties of such fluctuations. It has also been pointed out that the shape of this master curve suggests the existence of a region of thermodynamic instability of the solution against spontaneous demixing, suggesting the applicability of a simple, two-stage model of crystal nucleation. Here, at the first stage, concentration fluctuations generate abundant and relatively long-lived liquid regions where proteins cluster, and then, at the second stage, the rearrangement of clustered proteins into a crystalline form takes place within a characteristic time [278,279]. Importantly, the boundaries of the region with the universal features of critical fluctuations and crystal nucleation are defined by the presence of minor structural changes in a protein molecule, with such a link between conformation details and universal behavior being triggered by solvent-mediated interactions, and with abrupt changes in the values of solution spinodal temperatures *T_S_* being ascribed to a stepwise change in protein hydration and the related solution thermodynamics [279]. It has also been hypothesized that the correlation between the solution instability and structural changes in a protein undergoing crystallization can be explained by the inability of solvent-induced interactions to distinguish between residues belonging to the same or different proteins if they are located at comparable distances. As a result, at high enough protein concentrations, abrupt changes in the interprotein interactions that affect the solution stability can occur concurrently with the intraprotein interactions that contribute to the conformational stability and structural integrity of individual protein molecules [279].

### 5.2. Protein Amorphous Aggregation

In contrast to the processes described in the previous section that happen in supersaturated solutions of ordered proteins (such as crystallization and companion to it liquid–liquid phase separation, gelation, and aggregation), the process of amorphous aggregate formation represents one of the consequences of protein misfolding. Importantly, this form of misfolded protein aggregation is rather common, and can take place under particular conditions, even for amyloidogenic proteins when they fail to form amyloid fibrils. Similar to fibrillation (see next section), amorphous aggregate formation is driven by intermolecular interactions, and is linked to protein denaturation, since partially-folded conformations with exposed hydrophobic surfaces exhibit a greater propensity to aggregate [282]. In fact, multiple studies have clearly indicated that amorphous and fibrillar aggregation arises from the intermolecular association of partially-folded intermediates [282,283,284,285,286,287,288,289,290]. This requirement of the partial unfolding of a protein molecule that precedes intermolecular interactions links amorphous aggregates and amyloid fibril. Furthermore, being large, insoluble intermolecular ensembles, these misfolded aggregates are characterized by high levels of structural stability.

However, these two forms of misfolded aggregates are quite different on many levels. For example, fibrillar aggregates are highly ordered, β-sheeted ensembles, whereas in amorphous aggregation, proteins aggregate/oligomerize without forming specific high-order structures. Furthermore, besides the obvious morphological differences (fibrillar versus amorphous), amyloid fibrils and amorphous aggregates differ from each other by the mechanisms of their formation [291]. A typical fibrillation process is described by a sigmoidal curve characterized by the presence of a lag period that reflects the existence of a high free energy barrier associated with the nucleation of ordered structures of amyloid fibrils; this process can be accelerated by seeding (i.e., by the addition of the fragments of the preformed fibrils). The lag grows and can become huge in the presence of secondary nucleation, that is, the nucleation of branching or fragmentation of fibrils in addition to the primary nucleation of linear protofibrils [292,293]. In other words, the fibrils are formed via a nucleation and growth mechanism, where the overall reaction is rate-limited by the existence of a high free energy barrier associated with the nucleation. On the other hand, the formation of amorphous aggregates represents and instantaneous and spontaneous process that does not have a lag period and a high free energy barrier, and is not accelerated by seeding [291]. Since protein crystals and amyloid fibrils are formed by a nucleation and growth mechanism [294,295,296], and since crystallization and fibrillation can be accelerated by seeding [297,298], amyloid fibrillation is considered to be similar to crystallization [291,294,295,296]. On the other hand, amorphous aggregation was proposed to be analogous to the glass transition [291], with the glassy behavior of amorphous aggregates being reflected in the presence of heterogeneous conformations that are fixed by strong attractive forces producing various sites of interaction [291,299].

Importantly, not all amorphous aggregates are formed from denatured protein species. In fact, amorphous aggregation can also happen in the solutions of near-natively-folded proteins [300]. Examples of this phenomenon are given by the aggregation of the cataract-related P23T mutant of γD-crystallin [301] and bovine pancreatic trypsin inhibitor (BPTI) variant, BPTI-22, containing 22 alanines [302]. In the first case, an analysis of γD-crystallin aggregated under physiological conditions by solid-state NMR revealed the presence of a well-ordered, native-like conformation [301]. In the second case, the presence of a native-like structure in aggregated BPTI-22 was evidenced by ^15^N hetero single quantum correlation (HSQC) NMR spectra [302]. Since in many other cases, amorphous aggregates were formed from partially-folded intermediates, it is unclear now how widespread such near-natively-folded amorphous aggregates are.

Recently, it was pointed out that although amorphous aggregation and fibrillation—which can happen for a given protein at similar solution conditions—are often considered as competing processes, or amorphous aggregation is treated as an obligatory intermediate process within the amyloid formation pathway, these two models can be integrated into a single paradigm [303,304]. Here, amorphous aggregation is treated as a liquid–liquid phase transition leading to the formation of the amorphous aggregate that represents a second liquid phase whose liquid-like properties are determined by the intra-phase monomer mobility, and where fibrillation takes place at the interfacial boundary via the heterogeneous growth pathways including the nucleation, growth, and fragmentation of amyloids [292,293,303,304].

### 5.3. Protein Fibrillation

The interest of researchers in protein misfolding and fibrillation is based upon the involvement of these processes into the pathogenesis of various protein deposition diseases or proteinopathies, such as amyloidoses, and different neurodegenerative disorders. Globally, the molecular mechanisms underlining these different pathological states are the same, where first a transition of specific proteins or protein fragments from a native soluble form into insoluble aggregate/fibrils takes place, with the subsequent accumulation of aggregated material within a variety of organs and tissues [305,306,307,308,309,310,311,312]. Different proteinopathies are caused by unrelated proteins, which, prior to fibrillation, may be rich in α-helices, β-structure, or have an α/β or α + β structure. Some of these amyloidogenic proteins are globular proteins with unique 3D structures, whereas others are IDPs with different levels and degrees of disorder. Despite these differences, the fibrils found in different pathologies have many common features, such as similar morphologies, being twisted, rope-like structures, reflecting a filamentous substructure, and the presence of a core cross-β-sheet structure with continuous β-sheets formed by β-strands running perpendicular to the long axis of the fibrils [313]. Importantly, not all amyloids are pathological, and living organisms quite often exploit the inherent ability of proteins to fibrillate in order to generate functional amyloids with novel and diverse biological functionalities [314].

Since amyloid fibrils can be formed in vitro from both disease-associated and disease-unrelated proteins and peptides, it is now believed that the ability to fibrillate represents a generic property of a polypeptide chain, with all proteins being potentially able to form amyloid fibrils under the appropriate conditions [307,315,316,317,318,319]. Therefore, contrarily to the globular proteins that can spontaneously fold into unique 3D structures, which are critically dependent on the amino acid composition and sequence of the polypeptide chain [6,7], amyloid fibrils represent a generic phase of any peptide chain stabilized mostly via the main chain-main chain interactions, being therefore rather insensitive to the information encoded in the side chains.

These observations seem to suggest the absence of a specific “aggregation code” defining the formation of amyloid fibrils. However, not all proteins are equally prone to fibrillation; some are prone to aggregate, and can do so (if their concentration is sufficient) even under the physiological or near-physiological conditions, especially if the monomeric protein is deprived of its natural interacting partners, whereas others require rather extreme environmental perturbations (like seeding by fragments of preformed fibrils) to initiate the fibrillation process [307,311,316,320,321,322]. Furthermore, aggregation can be disallowed by some negative-design features present in the folded state of a protein [323]. Therefore, a selective pressure optimizes the primary sequence to allow a protein to fold into a stable soluble structure. This optimization is needed to prevent the functionally-competent fold from converting to the amyloid phase. Taken together, these findings suggest that not all soluble proteins have structures that are optimized to the same degree in order to avoid fibrillation, and that the susceptibility of a protein to conversion to the aggregated form is defined by the dependence of its structure on binding partnerships or complexation.

It is important to emphasize here that the formation of amyloid-like fibrils does not embody the only pathological feature of proteinopathies, and that pathological protein deposits can also be in the form of amorphous aggregates, which are cloud-like inclusions without defined morphologies. For many proteins, the aggregation process originating from protein misfolding can generate another alternative final product, i.e., soluble oligomers. It seems that the precursor of soluble aggregates is the most structured, whereas amyloid fibrils are formed from the least-ordered conformation via binding-induced folding. The choice between three pathogenic aggregation pathways, i.e., fibrillation, amorphous aggregation, and oligomerization, is determined by the peculiarities of the amino acid sequence, the protein concentration, and the environment.

#### 5.3.1. Conformational Prerequisites for Amyloidogenesis

The fibrillation of the majority of ordered proteins, which are not able to easily form amyloid fibrils under physiological conditions, requires denaturing conditions [307,311,316,320,321,322], suggesting that these proteins can fibrillate when their rigid native structure is destabilized and a partially-unfolded conformation is formed [282,306,307,308,309,310,311,312,316,320,321,324,325,326,327]. Obviously, such a requirement for partial unfolding is not applicable to IDPs with extended disorder, since they do not have stable and well-folded 3D structures in their native states. Instead, a primary step of the fibrillation of such proteins involves partial folding, leading to the stabilization of a partially-folded conformation [317,318,319,328,329,330]. Therefore, a general hypothesis of fibrillogenesis which is applicable to ordered proteins and IDPs suggests that protein fibrillation is critically dependent on the structural transformation of a native protein (ordered or intrinsically disordered) into a partially folded, aggregation-prone conformation, enabling the assembly of misfolded aggregates via specific intermolecular interactions of different physico-chemical nature, such as electrostatic attraction, hydrogen bonding, and hydrophobic interactions. Therefore, amyloid fibril formation is promoted when relatively unfolded protein species are formed under conditions whereby noncovalent interactions are still favorable.

#### 5.3.2. Fibrillogenesis of Globular Proteins Depends on Partial Unfolding

Significant evidence supports the idea that the fibrillogenesis of globular proteins requires their partial unfolding [282,306,307,308,309,310,311,312,316,320,321,324,325,326,327]. One should keep in mind that the unique 3D structures of ordered globular proteins under physiological conditions are not completely immobile, but have structural fluctuations of various degrees and timescales. Due to this conformational breathing, the structure of a globular protein represents a dynamic conformational ensemble containing tightly-folded species and multiple partially-unfolded conformations, with the former greatly predominating [331,332]. The native structures of globular proteins were shown to be destabilized by most mutations associated with the accelerated protein fibrillation and proteinopathies. As a result, these mutations caused the increase in the steady-state levels of partially-folded forms within the conformational ensemble of a mutated protein [306,307,312,316,320,324,326,333,334,335,336,337]. Conversely, the stabilization of a native protein structure via the specific binding of ligands or drugs can significantly reduce the amyloidogenicity of a protein [338,339,340,341,342,343,344,345,346]. Furthermore, the rate of fibril formation can be significantly accelerated by destabilizing the native structure of a globular protein by utilizing low or high pH, high temperatures, low to moderate concentrations of strong denaturants, organic solvents, etc.

#### 5.3.3. Fibrillogenesis of Extended IDPs Is Driven by Partial Folding

As mentioned, the functionalities of many proteins require a high degree of structural disorder [17,19,20,33,34,54,55,56,58,59,88,210,234,347]. It seems that due to their lack of significant conformational constraints and substantial conformational mobility, IDPs ought to be better suited to amyloidogenesis than to tightly-packed globular proteins; however, this is not always the case, and many extended IDPs are rather resilient against aggregation and fibrillation. This is because the fibrillation of such proteins requires their partial folding. Examples of such amyloidogenic extended IDPs undergoing partial folding during their amyloidogenesis are Aβ [328], tau protein [348], α-synuclein [317], β-synuclein [349], γ-synuclein [350], amylin [351], prothymosin α [352], and histones [319].

#### 5.3.4. Premolten Globule as a Universal Amyloidogenic Intermediate

Based on the analysis of structural events at the early fibrillation stages, it has been concluded that the substantially unfolded conformations of proteins and polypeptides typically serve as fibril precursors [311]. Although any partially-folded species (including the molten globular and the premolten globular species) may potentially play a role of such a crucial fibrillation-prone intermediate, the accumulated evidence indicates that the amyloidogenic species is significantly unfolded, being structurally closer to the premolten globule than to the molten globule state [311]. It seems that among different partially-folded intermediates described for proteins, the most amyloidogenic species is the premolten globule state, which is a relatively swollen conformation lacking a globular structure and possessing a relatively low secondary structure content, i.e., that sums to ~50% or less of the corresponding native value [17].

#### 5.3.5. Sequential Mechanism of Fibril Formation and Morphological Heterogeneity of Amyloid Fibrils

In conclusion, proteins with different types of structures are equally subjected to aggregation [311,354], which represents an extremely complex process consisting of at least three major steps. First, different soluble proteins are transformed into the “sticky” aggregation-prone precursor or intermediate with the premolten globule properties. Since such aggregation-prone intermediates would be structurally different for different proteins, and since even the same protein can be converted into the structurally-different, partially-folded species by different environmental conditions, the variations in the amount of the ordered structure retained in the amyloidogenic precursor is believed to be responsible for the formation of fibrils with distinct morphologies [355]. At the second step, which is usually considered a nucleation step, or the lag period that precedes the formation of the insoluble aggregates, different oligomeric species are formed [354]. The lag phase occurs because the association of monomers is initially unfavorable, however, once a critical nucleus has been generated, and the gain of enthalpy from incorporating additional monomers outweighs the increase in entropy from dissociation. As a result, aggregation becomes energetically favorable and the reaction enters a growth phase [354]. An idealized model of amyloid fibril formation and protein aggregation in general is presented in Figure 7 (see bottom pathway), illustrating the directionality and sequential nature of the aggregation process that includes a series of consecutive steps [353]. Importantly, not all oligomers formed during the protein fibrillation process are productive, i.e., not all of them will eventually “grow” into fibrils. In fact, some metastable oligomers can “crouch”, being able to compete with fibril formation by decreasing the concentration of the fibril-forming free monomers [293]. This scenario is illustrated by the presence of competing misfolded oligomers in the bottom panel of Figure 7. Depending on the peculiarities of the environment, different aggregated forms (oligomers, amyloid fibrils, amorphous aggregates) can be generated via the intermolecular self-assembly of the different partially-folded species of a given protein (see upper pathways in Figure 7). In this model, different oligomers are formed from the structurally-identical monomers. However, since aggregation can cause dramatic structural reorganization of the aggregating protein, monomers at different aggregation stages are not structurally identical [353]. Again, one can expect to find competing misfolded oligomers within all the pathways shown in Figure 7.

Furthermore, the typical aggregation process only rarely results in the formation of a homogeneous intermolecular ensemble, where only one type of aggregates species is present. More often, various aggregated forms appear, giving rise to heterogeneous mixtures of differently-aggregated species (see Figure 8). In addition, for each aggregated form, there is a multitude of different morphologies, with monomers constituting such morphologically-distinctive aggregated forms being potentially structurally diverse (see Figure 7). All of this suggests that far from being a simple reaction, aggregation represents a very complicated process with multiple related and unrelated pathways, which can be connected or disjoined.

## 6. Reincarnation of Liquid–Liquid and Liquid–Gel Phase Transitions: Drivers of the Biogenesis of Membraneless Organelles

Although the phenomenon of liquid–liquid phase separation (LLPS) in supersaturated protein solutions has been known in the field of protein crystallography for a long time, it was mostly unknown for the outside world. The situation has changed recently, and we are now witnessing a dramatic increase in the level of interest in this intriguing phenomenon, not only from crystallographers, but also among researchers working in various fields of protein science, cellular biology, biotechnology, and biomedicine. This is because of the realization that LLPS can drive the cellular compartmentalization and biogenesis of various membraneless organelles (also known as the proteinaceous membraneless organelles (PMLOs), puncta, “biomolecular condensates”, foci, etc.). Curiously, although the existence of such membraneless compartments within the cells has been known to the scientific community for many years (e.g., nucleolus was described as early as in the 1830s [356,357]), the facts that PMLOs are numerous, and that they may have important biological functions, were generally overlooked, mostly due to the inability to isolate them for focused analyses which were in line with a scientific reductionistic approach, i.e., if the functionality of a complex system is the sum of the functions of its constituents, then to understand how such a complex system works, it needs to be taken apart, and individual parts need to be studied separately to understand their structures and functionalities. Although this linear scientific method was successfully utilized for the analysis of the functionality of “traditional” membrane-encapsulated organelles, it obviously failed for PMLOs (no membrane equals no luck with isolation). As a result, for a long time, this inability to be isolated, combined with their transient existence, have placed PMLOs in the category of potential artifacts.

It is recognized now that PMLOs, these highly dynamic protein-based assemblages [358], are often present in cytoplasms, nuclei, the mitochondria of various eukaryotic cells, in chloroplasts of plant cells, as well as in bacterial cells, where they play a number of important roles in the organization of various intracellular processes [358,359,360,361,362,363,364]. PMLOs are numerous and very diverse [359,361,365,366,367,368,369,370,371,372]; there are at least 40 different types found in eukaryotic and bacterial cells [373]. Figure 9 illustrates the multiplicity of PMLOs by showing a horde of such phase-separated liquid droplets that can be found in bacterial and eukaryotic cells. A detailed description of eukaryotic PMLOs with illustrative examples is beyond the scopes of this article, and is presented elsewhere [374].

The formation of PMLOs represents a natural way of compartmentalizing various biological processes in different regions of the cell [360]. Since PMLOs are able to respond to, facilitate, regulate, and control different biological functions and stimuli [361], they are now considered to be important controllers of cellular life.

The liquid-like nature of PMLOs and phase-separated droplets can affect and modulate the functions of their components, which remain dynamic and flexible within these droplets, despite being amassed at high concentrations. In line with these considerations, it has been shown that the low-density structure of PMLOs found within the *Xenopus* oocyte nuclei determines the access from the nucleoplasm to the macromolecules within these PMLOs [375]. Due to their increased concentrations of nucleic acids and proteins, PMLOs can accelerate cytoplasmic reactions, thereby behaving as liquid-phase microreactors [376,377,378]. They can also represent a way recruiting and concentrating specific proteins, as, for example, observed in Negri bodies (NBs), where viral RNAs are synthesized [379]. Since some nuclear PMLOs concentrate specific sets of mRNAs and regulatory proteins, they also can serve as dynamic sensors of localized signals and, thereby, play a dual role in the translation of associated mRNAs, preventing mRNA translation at rest, and promoting local protein synthesis upon activation [380].

There is no doubt that PMLOs are full of mystery. Since they do not have membranes, their biogenesis and structural integrity rely exclusively on protein–protein and/or protein–nucleic acid interactions [381,382], and their components can directly contact and exchange with the exterior environment [383,384]. On the other hand, they have macroscopic dimensions and are detectable by under a microscope. The dimensions of these highly mobile but stable assemblages are dependent on the cell size [376]. PMLOs demonstrate the liquid-like behavior, being capable of dripping, the formation of spherical structures upon fusion, and wetting [385,386,387,388].

These intracellular liquid droplets are formed via biological, liquid–liquid phase transitions (LLPTs), also known as intracellular liquid–liquid demixing phase separation [376,389]. Such intracellular LLPTs are concentration-dependent, since PMLO formation is initiated by the colocalization of the participating molecules at high concentrations within a small cellular microdomain [383,384]. The biogenesis of PMLOs is a highly controllable and reversible process, with the formation of PMLOs being initiated by fluctuations in the concentrations of proteins undergoing LLPT, variations in the concentrations of definite small molecules or salts, osmolarity changes, alterations in the solution pH and/or temperature, by alternative splicing and various PTMs of the phase-forming proteins, via binding of these proteins to some specific partners, or by alterations of other environmental conditions affecting protein–protein or protein–nucleic acid interactions [376,389,390,391,392].

The fluidity of PMLOs originates from the multivalent interactions between IDPs or proteins containing IDPRs that are not accompanied by noticeable alterations in the structure of proteins undergoing LLPTs [372,393,394]. Therefore, PMLOs represent a special form of disorder-based protein complexes [372,389,393,395], and can be considered as illustrations of the disorder-based emergent behavior of proteins [210,213,396,397]. The lack of noticeable structural changes in IDPs forming PMLOs is supported by the NMR analysis of several PMLOs or liquid droplets, such as the Alzheimer-related protein tau [398,399], elastin-like polypeptides (ELPs) [400], low-complexity domain of the RNA-binding protein fused in sarcoma (FUS) [401], and the heterogeneous nuclear ribonucleoprotein A2 (hnRNPA2) [402], to name a few.

## 7. Conclusions

This review describes various intra- and inter- molecular phase transitions taking place in protein solutions, thereby representing protein existence as an exciting story of life in phases, where different phase transitions define the structure, function, interactability, aggregation, crystallization, and compartmentalization of proteins. Although many of these phase transitions are linked to the general polymeric nature of proteins (e.g., intramolecular coil-globule transitions or intermolecular liquid–liquid and liquid–gel phase separation), other phase transitions seems to be rather specific for proteins, which are biological copolymers that were evolutionarily edited to have unique structures and/or functions. This edited polymer nature is related to the ability of globular proteins to undergo intramolecular, globule–globule transitions, giving rise to their unique 3D structures and their ability to be crystallized, as well as being associated with the ability of intrinsically-disordered proteins to undergo binding-induced intramolecular phase transitions originating from their interactions with specific partners. Also, although it seems that amorphous aggregation can take place in supersaturated solutions of various solutes of polymeric and nonpolymeric nature, ordered aggregations, i.e., the formation of amyloid fibrils that requires a dramatic structural rearrangement of protein monomers, might represent a special case of intermolecular phase transitions which is specific to polypeptides.

## Figures and Tables

**Figure 1 biomolecules-09-00842-f001:**
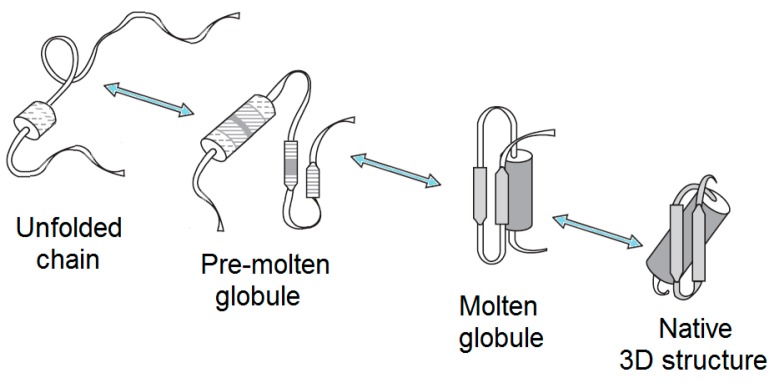
Four main possible stable states of protein molecule: noncompact unfolded chain with, maybe, some traces of secondary structures; swollen “premolten” globule with partly formed secondary structures; compact “molten globule” with almost formed secondary structures and folding pattern, but having no close packing of its mobile side chains; and solid native protein structure [22].

**Figure 2 biomolecules-09-00842-f002:**
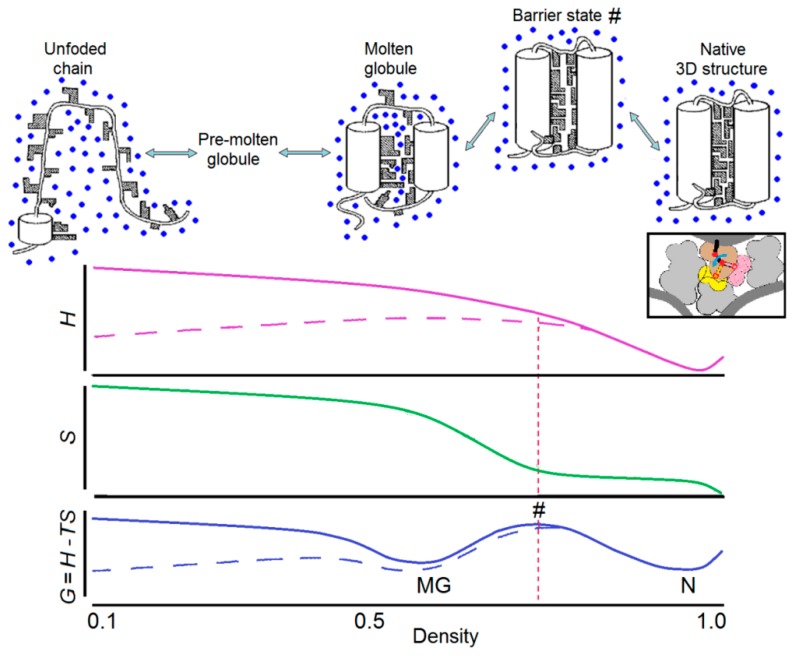
At the top of the picture: The main possible stable states of the protein molecule and the unstable barrier state providing the all-or-none transition between the native structure (N) and the molten globule (MG), and all of the unfolded forms. Blue dots indicate water molecules. Inset: A sketch of a small piece of the close side chain packing. The yellow side chain “head” corresponds to an alternative rotamer of the central side chain, which is forbidden by close packing. At the bottom of the picture: Enthalpy *H*, entropy *S*, and free energy *G* of the protein molecule, depending on its uniform density. *T* is the temperature of the MG ↔ N equilibrium in a “bad” solvent. The dashed lines correspond to a “better” solvent. As is customary in the literature on protein folding theory, the “entropy” *S* does not include the solvent entropy; correspondingly, “enthalpy” *H* means, actually, the “free energy of interactions” (also called the “mean force potential”), since, e.g., the hydrophobic, electrostatic and other solvent-mediated forces, with all their solvent entropy, are included in this “enthalpy”. Adapted from [1,153].

**Figure 3 biomolecules-09-00842-f003:**
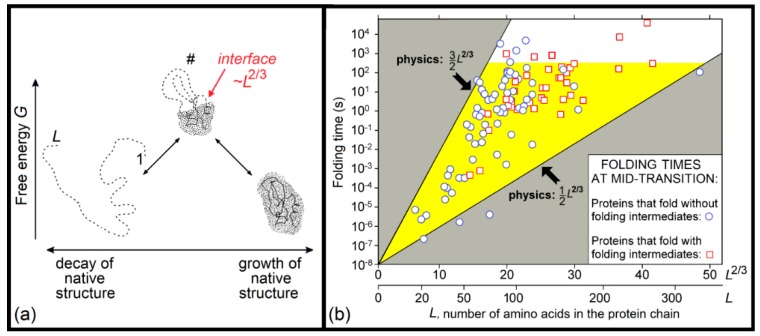
(**a**) A scheme of the reversible “all-or-none” transition from the unfolded chain to the native globular structure; # marks the rate-determining transition state whose free energy is proportional to the size of the maximal interface of the native and unfolded phases, which scales with the chain length *L* as *L*^2/3^. (**b**) Experimentally-measured in vitro folding times at N ↔ U equilibrium for 107 single-domain proteins (or separate domains) without SS bonds and covalently bound ligands (although the folding rates for proteins with and without SS bonds are principally the same [163]). Triangle: the region allowed by physics; its golden part corresponds to biologically-reasonable folding times (≤10 min) under “biological” ambient conditions; the larger folding times (in the white zone) are observed (for some proteins) only under the equilibrium, i.e., nonbiological conditions. Adapted from [1,176].

**Figure 4 biomolecules-09-00842-f004:**
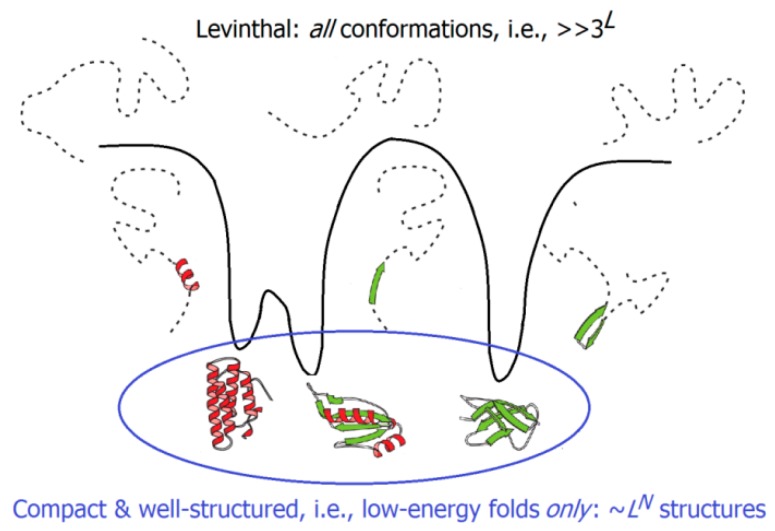
Comparison of a huge search among *all*, for the most part disordered conformations, and a much less voluminous search among *only* compact and well-structured globules, which correspond to the deep energy minima. *L* is the number of amino acid residues in the protein chain, and *N* is the number of secondary structure elements in this chain, which can be estimated as *N* ≈ *L*/[(3 ÷ 4.5)*L*^1/3^] ≈ *L*^2/3^/4 [181].

**Figure 5 biomolecules-09-00842-f005:**
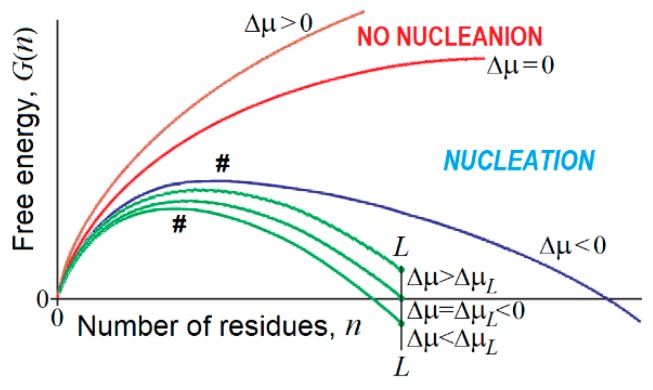
Free energy of a growing piece of a “new” phase at different Δμ values. The symbol # shows the transition state of the process. Three short green lines correspond to “all-or-none” transitions within a protein-like body formed by an *L*-residue chain (which can be in two phase states) at the equilibrium point (Δμ = Δμ_L_, see the text), as well as and somewhat above and below of this point.

**Figure 6 biomolecules-09-00842-f006:**
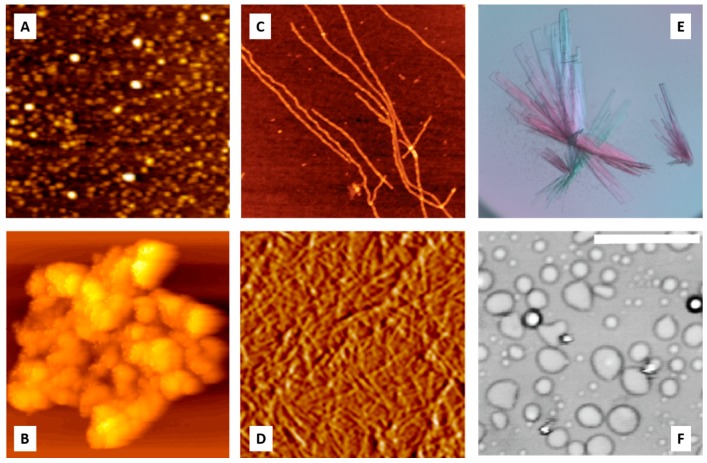
Illustrative examples of phases formed in protein solutions as a result of intermolecular phase transitions: (**A**). Oligomeric species; (**B**). Amorphous aggregate; (**C**). Amyloid fibrils; (**D**). Protein gel (atomic force microscopy (AFM) image of lysozyme fiber gel); (**E**). Protein crystal (crystals of slingshot phosphatase 2 in a hanging drop at 20 °C); (**F**). Liquid droplets (liquid droplets formed by the C-terminal domain of TDP-43 in the presence of RNA as a result of LLPT. Here, 20 μM TDP-43CTD in presence of 40 μg/mL RNA forms liquid droplets that coalesce over time. Scale bar represents 20 μm). Images in plots (**A**–**C**) showing different aggregated forms of α-synuclein are AFM images from a personal collection of V.N.U. Image in plot (**D**) is modified with permission from [269]. Image in plot (**E**) is a courtesy of Dr. Eric M. Lewandowski and Prof. Yu Chen, University of South Florida. Image shown in plot (**F**) is a courtesy of Mr. Anukool A. Bhopatkar and Prof. Vijayaraghavan Rangachari, University of Southern Mississippi.

**Figure 7 biomolecules-09-00842-f007:**
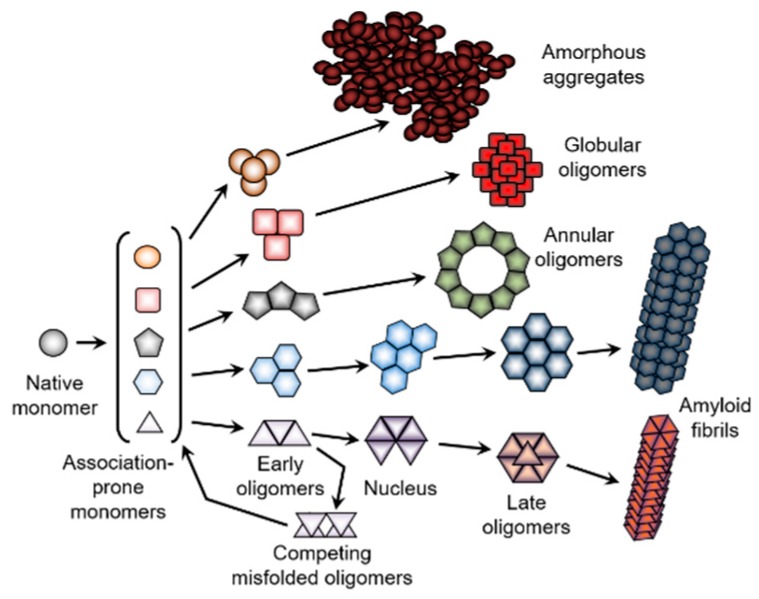
Schematic, oversimplified depiction of the process of protein self-association. Multiple aggregation pathways are generated via the formation of multiple association-prone monomeric forms. The aggregation reaction generates at least three major products, which are amorphous aggregates (top pathway), different soluble oligomers with diverse morphologies (second and third from the top pathways), and morphologically-divirgent amyloid fibrils (two bottom pathways). Potential structural changes in the monomers that might happen at each elementary step are shown by color changes. The real situation is more complex, and more different species can be formed during and as a result of aggregation. Various species within the different pathways can interconvert. Modified from [353].

**Figure 8 biomolecules-09-00842-f008:**
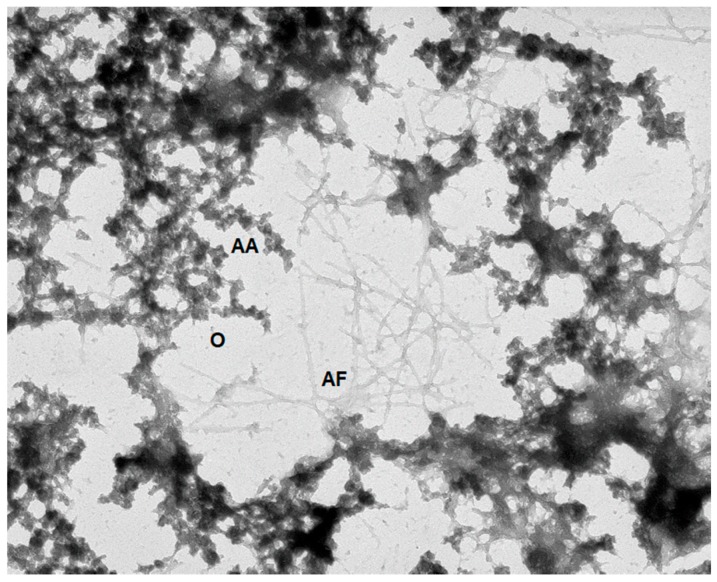
Transmission electron micrograph of a typical end-product of the protein aggregation process (in this case, fibrillation of α-synuclein), where amorphous aggregates (AA), amyloid-like fibrils (AF), and oligomers (O) are present. This image is from the personal collection of V.N.U.

**Figure 9 biomolecules-09-00842-f009:**
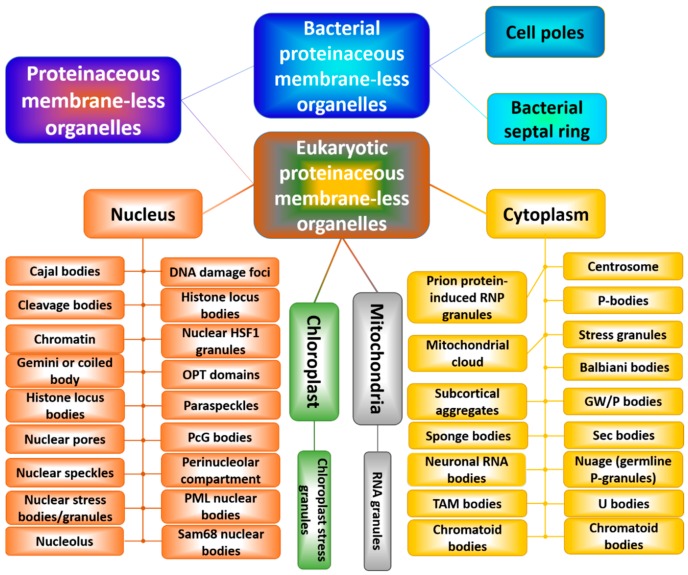
Illustration showing the variability of chloroplast, cytoplasmic, mitochondrial, and nuclear, proteinaceous membraneless organelles (PMLOs) that can be found in eukaryotes and bacterial PMLOs. This figure was used with permission from Zaslavsky, B.Y., and Uversky, V.N. (2018). In Aqua Veritas: The Indispensable yet Mostly Ignored Role of Water in Phase Separation and Membraneless Organelles. Biochemistry 57(17), 2437–2451. Copyright (2018) American Chemical Society.

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
