# Peer review of "Life in Phases: Intra- and Inter- Molecular Phase Transitions in Protein Solutions"

_biomolecules, 2019, doi:10.3390/biom9120842_

Round 1

Reviewer 1 Report

Excellent and very comprehensive review concerning the characteristics and mechanisms involved in intra-molecular (protein folding) and multiple cases of inter-molecular phase transitions. The authors reviewed vast material on this subject including their own substantial contributions to these fields. The review is very timely, clearly written, the reviewed literature list is very extensive (ca. 400 articles), the figures are demonstrative and add to the clarity of overall presentation. The review is highly recommended for publications as of great value and interest to a very broad readership and without any further reservations.

Author Response

We are thankful to this reviewer for high evaluation of our work.  

Reviewer 2 Report

The review by Uversky et al is an exceptional work linking multiple areas of biophysics together under a single framework. I only have one minor suggestion, which is completely optional:

Lines 290-297:

“Thus, a small expansion of the native protein globule is always unfavorable, because it already increases the globule’s energy (whose parts already lose their close packing), but does not yet increase the globule’s entropy (since it does not yet liberate the rotational isomerization of the side  groups). That is, the globule’s free energy always increases with a small expansion. As a result, protein denaturation occurs not gradually, but as a jump over the free energy barrier, and this leads to the “all-or-none” kind of transition (Figure 2).

The aforementioned mechanism is related to the transition of a native globular state to any denatured form: molten globule, or pre-molten globule, or coil. Therefore, the protein structure tolerates, without a change, a change of ambient conditions up to a certain limit, and then melts as a  whole, like a macroscopic crystal.

This explanation is perhaps slightly imprecise. I would suggest the following small changes (changes highlighted in bold) to highlight the conditions in which this explanation holds. I would also suggest adding the reference indicated below for the expression of (dE/dV)T in terms of the experimentally measurable coefficients of thermal expansion and compressability:

“Thus, a small expansion of a compact native protein globule is always slightly unfavorable,[ref 153] because it already increases the globule’s energy (whose parts already lose their close packing), but does not yet increase the globule’s entropy (since it does not yet liberate the rotational isomerization of the side groups or allow entry of water into the protein core). That is, the globule’s free energy always increases with a small expansion.1 As a result, transitions between states during protein denaturation occur not gradually, but as a jump over the free energy barrier, and this leads to the “all-or-none” kind of transition (Figure 2).

The aforementioned mechanism is related to the transition of the native globular state to any denatured form: molten globule, or pre-molten globule, or coil.[refs 141 and 152] Therefore, the native protein structure tolerates, without significant change, a change of ambient conditions up to a certain limit, and then melts as a  whole, like a macroscopic crystal.

            (1)        Rees, D. C.; Robertson, A. D. Some thermodynamic implications for the thermostability of proteins. Protein Sci 2001, 10, 1187-1194.

Author Response

The review by Uversky et al is an exceptional work linking multiple areas of biophysics together under a single framework.

REPLY: We are thankful to this reviewer for high evaluation of our work.  

I only have one minor suggestion, which is completely optional:

Lines 290-297:

“Thus, a small expansion of the native protein globule is always unfavorable, because it already increases the globule’s energy (whose parts already lose their close packing), but does not yet increase the globule’s entropy (since it does not yet liberate the rotational isomerization of the side  groups). That is, the globule’s free energy always increases with a small expansion. As a result, protein denaturation occurs not gradually, but as a jump over the free energy barrier, and this leads to the “all-or-none” kind of transition (Figure 2).

The aforementioned mechanism is related to the transition of a native globular state to any denatured form: molten globule, or pre-molten globule, or coil. Therefore, the protein structure tolerates, without a change, a change of ambient conditions up to a certain limit, and then melts as a  whole, like a macroscopic crystal.”

This explanation is perhaps slightly imprecise. I would suggest the following small changes (changes highlighted in bold) to highlight the conditions in which this explanation holds. I would also suggest adding the reference indicated below for the expression of (dE/dV)T in terms of the experimentally measurable coefficients of thermal expansion and compressability:

“Thus, a small expansion of a compact native protein globule is always slightly unfavorable,[ref 153] because it already increases the globule’s energy (whose parts already lose their close packing), but does not yet increase the globule’s entropy (since it does not yet liberate the rotational isomerization of the side groups or allow entry of water into the protein core). That is, the globule’s free energy always increases with a small expansion.1 As a result, transitions between states during protein denaturation occur not gradually, but as a jump over the free energy barrier, and this leads to the “all-or-none” kind of transition (Figure 2).

The aforementioned mechanism is related to the transition of the native globular state to any denatured form: molten globule, or pre-molten globule, or coil.[refs 141 and 152] Therefore, the native protein structure tolerates, without significant change, a change of ambient conditions up to a certain limit, and then melts as a  whole, like a macroscopic crystal.”

            (1)        Rees, D. C.; Robertson, A. D. Some thermodynamic implications for the thermostability of proteins. Protein Sci 2001, 10, 1187-1194.

REPLY: Thank you for pointing this out. We accepted almost all proposed changes except the little things and the proposed link to Rees at al., since this work refers to the stable expansion of the protein, and not to climbing onto the barrier (for the same reason, we did not accept “slightly unfavorable”). 

This section of the manuscript now reads:

Thus, a small expansion of the compact native protein globule is always unfavorable [152. 153], because it already increases the globule’s energy (whose parts already lose their close packing), but does not yet increase the globule’s entropy (since it does not yet liberate the rotational isomerization of the side groups) or allow entry of water into the protein core. That is, the globule’s free energy always increases with a small expansion. On the contrary, a large globule’s expansion liberates the rotational isomerization of the side groups and leads (at high enough temperature) to decrease of the free energy. As a result, protein denaturation occurs not gradually, but as a jump over the free energy barrier, and this leads to the “all-or-none” kind of transition (Figure 2).

The aforementioned mechanism is related to the transition of a native globular state to any denatured form: molten globule, or pre-molten globule, or coil [141. 152]. Therefore, the protein structure tolerates, without significant change, a change of ambient conditions up to a certain limit, and then melts as a whole, like a macroscopic crystal.

Reviewer 3 Report

This is a really nice review integrating different types of phase transitions of proteins. I only have minor comments detailed below.

The first paragraph is too long. It might be good to split into two or three: energies in different environments, possible variation of environments and corresponding protein structure variation.

Fig 1, 'Unfoded' should read 'Unfolded'.

L101, it might be better to start a new paragraph to talk about function of IDPs.

L174, there is a gap between introduction of polypeptide phase transition and mechanism of IDP against cellular degradation. There are also several types of phase transitions the authors condense in a few lines of this paragraph. It might help by incorporating different types of phase transitions into the states in Fig. 1, or into a separate schematic diagram after reading section 2.

L213, 'mas' -> 'mass'.

L225, it's a little bit confusing to read 'pre-molten globule'. A few more sentences might be added to explain its difference to the unfolded state, since unfolded state also contains structures and is not fully random coil for an IDP.

L313, two 'does not'.

L357-361, there is also an analytical version (PNAS 89,20, 1992).

L466, 'unfoldon' seems to be a new term and the authors might want to explain more. This is also a long paragraph difficult to follow, starting with 'foldon' of folded proteins and getting to heterogeneity of IDP conformations.

L572, there might be a way to put simple figures for Eq. 5, which helps illustrate all the observations.

L637, the title of this section (about behavior in supersaturated solutions) might not fit exactly the content since aggregation, gelation and liquid-liquid phase separation are also discussed.

L677, it might help by showing a schematic phase diagram.

L810, the section ID is not correct starting from here.

L992, there is an increasing interest on the relation between these droplets and other phases like amorphous aggregates and amyloid fibrils. Not sure if the authors would be interested in adding one paragraph here.

The authors seem to have the perspective that polymeric nature of the proteins instead of protein sequences control many of these phase transitions. Not sure if they want to add comments that there can also be concerns of sequence determinants.

Author Response

This is a really nice review integrating different types of phase transitions of proteins. I only have minor comments detailed below.

REPLY: We are thankful to this reviewer for high evaluation of our work and useful comments and suggestions.  

The first paragraph is too long. It might be good to split into two or three: energies in different environments, possible variation of environments and corresponding protein structure variation.

REPLY: Thank you for pointing this out. First paragraph was split into two as recommended

Fig 1, 'Unfoded' should read 'Unfolded'.

REPLY: Corrected

L101, it might be better to start a new paragraph to talk about function of IDPs.

REPLY: Thank you for pointing this out. A new paragraph was started to talk about function of IDPs.

L174, there is a gap between introduction of polypeptide phase transition and mechanism of IDP against cellular degradation. There are also several types of phase transitions the authors condense in a few lines of this paragraph. It might help by incorporating different types of phase transitions into the states in Fig. 1, or into a separate schematic diagram after reading section 2.

REPLY: Thank you for pointing this out. However, we do not think that the mentioned gap should be filled. We also do not think that proposed changes are needed. In our view, This section of the manuscript is clear.

L213, 'mas' -> 'mass'.

REPLY: Corrected

L225, it's a little bit confusing to read 'pre-molten globule'. A few more sentences might be added to explain its difference to the unfolded state, since unfolded state also contains structures and is not fully random coil for an IDP.

REPLY: Thank you for pointing this out. This sentence was changed to read: “Finally, it has been shown that the pre-molten globule (with relatively large secondary structure content), as well as the unfolded state (with low content of residual secondary structure), is separated from the molten globule by a sharp transition, which, in some proteins, represents an "all-or-none" transition; i.e., an intramolecular analog of the first order phase transition [11,13-15,17,116,141].”

L313, two 'does not'.

REPLY: Corrected

L357-361, there is also an analytical version (PNAS 89,20, 1992).

REPLY: Thank you for pointing this out. This sentence was changed to read: “In the line with the earlier analytical estimates {Zwanzig, 1992 #425}, computer experiments have shown that a model polymer, whose random sequence was slightly "edited" to make free energy of its most stable fold lower than that of any other fold by at least by a few kcal/mol [168,169] finds this most stable fold in a time, which is by many orders of magnitude smaller than the time necessary to iterate over all possible chain structures.”

L466, 'unfoldon' seems to be a new term and the authors might want to explain more. This is also a long paragraph difficult to follow, starting with 'foldon' of folded proteins and getting to heterogeneity of IDP conformations.

REPLY: Thank you for pointing this out. Some clarification was added here “On the other hand, functionality of many ordered proteins depends on the presence of 'unfoldons’; i.e., regions of ordered proteins undergoing that undergo order-to-disorder transition to make protein active [209].”

L572, there might be a way to put simple figures for Eq. 5, which helps illustrate all the observations.

REPLY: Done

L637, the title of this section (about behavior in supersaturated solutions) might not fit exactly the content since aggregation, gelation and liquid-liquid phase separation are also discussed.

REPLY: Thank you for pointing this out. The title of this section was changed to read “Protein crystallization as a peculiar case of phase separation of supersaturated protein solutions”

L677, it might help by showing a schematic phase diagram.

REPLY: Thank you for pointing this out. However, we do not think that such a diagram is needed

L810, the section ID is not correct starting from here.

REPLY: Thank you for pointing this out. Numbering of corresponding sections were changed

L992, there is an increasing interest on the relation between these droplets and other phases like amorphous aggregates and amyloid fibrils. Not sure if the authors would be interested in adding one paragraph here.

REPLY: Thank you for pointing this out. However, we do not think that addition of such a paragraph is required here.

The authors seem to have the perspective that polymeric nature of the proteins instead of protein sequences control many of these phase transitions. Not sure if they want to add comments that there can also be concerns of sequence determinants.

REPLY: Thank you for pointing this out. However, we do not think that addition of such comments is required here.

This manuscript is a resubmission of an earlier submission. The following is a list of the peer review reports and author responses from that submission.